# Are Language Models Worse than Humans at Following Prompts? It's Complicated

Albert Webson[κφ*], Alyssa Marie Loo[κλ*], Qinan Yu[κ*], and Ellie Pavlick[κλ]

[κ]Department of Computer Science    [φ]Department of Philosophy
[λ]Program in Linguistics

Brown University

## Abstract

Prompts have been the center of progress in advancing language models' zero-shot and few-shot performance. However, recent work finds that models can perform surprisingly well when given intentionally irrelevant or misleading prompts. Such results may be interpreted as evidence that model behavior is not "human like". In this study, we challenge a central assumption in such work: that humans would perform badly when given pathological instructions. We find that humans are able to reliably ignore irrelevant instructions and thus, like models, perform well on the underlying task despite an apparent lack of signal regarding the task they are being asked to do. However, when given deliberately misleading instructions, humans follow the instructions faithfully, whereas models do not. Our findings caution that future research should not idealize human behaviors as a monolith and should not train or evaluate models to mimic assumptions about these behaviors without first validating humans' behaviors empirically.

## 1   Introduction

Prompting has emerged as the default way of using large language models (Brown et al., 2020; Sanh et al., 2022; Wei et al., 2021; Ouyang et al., 2022; Chung et al., 2022). However, a collection of recent papers show that models perform surprisingly well when given misleading or irrelevant prompts (Webson and Pavlick, 2022; Prasad et al., 2022; Khashabi et al., 2021), corrupted in-context examples (Min et al., 2022), or corrupted explanations or chain-of-thought (Madaan and Yazdanbakhsh, 2022; Ye and Durrett, 2022). Such results raise questions about whether language models' ability to follow instructions is analogous to humans' ability to do so.

In this paper, we investigate what humans do in such settings. We follow the experimental setup used by Webson and Pavlick (2022) (W&P hereafter) to study how instructions in prompts affect LMs' performance. W&P manually write a set of prompts for various natural language inference (NLI) and coreference resolution datasets. These prompts cover three main categories: instructive, irrelevant, and misleading. For example, `Given {sentence1} , can we assume it is true that {sentence2}` is an instructive prompt for NLI, whereas `{sentence1} Does the above passage express a positive sentiment? {sentence2}` is a misleading instruction for NLI. (See Table 1 for full definitions and examples.)

W&P assume that, if models perform held-out tasks by reading prompts as instructions in the way that humans (are assumed to) do, their performance with instructive prompts should be much higher than their performance with other pathological categories of prompts, namely:

$$\text{instructive} > \text{misleading} \qquad \text{(A1)}$$
$$\text{instructive} > \text{irrelevant} \qquad \text{(A2)}$$
$$\text{instructive} > \text{null (no instruction)} \qquad \text{(A3)}$$

Instead, W&P find that T5 (LM-Adapted, 11B, Lester et al., 2021), T0 (11B, Sanh et al., 2022), and GPT-3 (175B, Brown et al., 2020) do not exhibit the above patterns. Rather, in both zero-shot and few-shot settings, models perform roughly the same on instructive, misleading, and irrelevant prompts, violating A1 and A2 above. Models do, however, perform better given any type of instructions than they do with no instructions (i.e., A3 holds). Therefore, W&P conclude that while prompts do confer substantial empirical benefits, the fact that models are so good at inferring the gold labels under various pathological prompts casts doubts on whether models understand or use instructions in ways similar to how humans do.

In this paper, we revisit W&P's assumptions on how humans behave with pathological prompts.

---

*Equal contribution. Correspondence to papers@aw.nyc.

| Category | Description | Examples |
|---|---|---|
| instructive | How we would describe the NLI task to a human who has never seen the task before. | {sentence1} Are we justified in saying that "{sentence2}"? 
 Given {sentence1} Should we assume that "{sentence2}" is true? |
| misleading | Instruct the models to perform a task unrelated to NLI. | {sentence1} is the sentiment positive? {sentence2} 
 {sentence1} is this a sports news? {sentence2} |
| irrelevant | Concatenate the premise, a sentence unrelated to any NLP task, and the hypothesis. | {sentence1} If bonito flakes boil more than a few seconds the stock becomes too strong. "{sentence2}"? |
| null | Concatenate the premise and the hypothesis without any additional text. | {premise} {hypothesis} 
 {sentence2}{sentence1} |

Table 1: Prompt categories adapted from W&P, with W&P's two misleading categories collapsed into one 'misleading' category for clarity. See Table 3 for the full list.

We use the same experimental design but adapt it for measuring human behaviors. In the zero-shot setting, we find that while assumptions A1 and A3 are consistent with human behaviors, A2 is not.

Our experiments underscore the importance of validating assumptions about human behavior on natural language tasks since, frequently, researchers' intuitions about human behavior do not bear out in practice, and that extra care should be taken in designing a fair comparison between models and humans (Pavlick and Kwiatkowski, 2019; Dasgupta et al., 2022; Lampinen, 2022).

## 2 Experiment

**Overview**    Following W&P, we use natural language inference (NLI) as the primary task for our experiments. (That is, in our results, we always report human and model performance with respect to the NLI task.) When necessary, in designing stimuli for the misleading prompt condition (discussed in detail below), we additionally draw examples from 7 other tasks: lexical overlap, lexical identity, paraphrasing identification, grammatical acceptability, summarization acceptability, topic classification, and language identification (see Appendices H.2 and H.3 for details); we refer to these collectively as *surface tasks* in this paper.

We define an *example* to be a pair of `<sentence1, sentence2>`. For NLI, sentences 1 and 2 are the premise and hypothesis, respectively.[1] We define an *item* as a unique 3-tuple `<sentence1, instruction, sentence2>`, i.e., an example fitted within a prompt template, which can be instructive (w.r.t. NLI), misleading, irrelevant, or empty (null).[2]

Crucially, when the instruction is misleading, we manually select examples such that `<sent1, misleading instruction, sent2>` and `<sent1, NLI instruction, sent2>` always have opposite gold labels—see Figure 1. Assuming participants are competent at NLI as well as at each of the relevant surface tasks[3] used in our experiments, this design enables us to distinguish whether the participant is performing the NLI task or surface task when given misleading instructions.

**Procedure**    To ensure this experiment is as zero-shot as possible, each participant receives only one test item, followed by four additional items which we use as controls to ensure that all tasks and examples are fair (Appendix E). For each item, subjects choose between "Yes" or "No". They do not receive any feedback throughout the experiment.

**Example Selection**    Because our main goal is to measure the effect of instructions and not humans' performance on the tasks themselves per se, we manually select examples that are as easy and unambiguous as possible. For the instructive, irrelevant, and null conditions, we choose examples from RTE (Dagan et al., 2006) and MNLI (Williams et al., 2018). For the misleading condition, we manually select all examples to ensure that the NLI labels differ from the misleading task labels. A full description of how we curate examples is detailed in Appendix H.

**Instruction Templates**    We select and lightly edit[4] 22 of the 27 prompts used in W&P for testing. The complete prompts are listed in Appendix H. Combined with examples, there are a total of 194 unique items in our test condition (Table 2). Each item is assigned to a minimum of three annotators.

**Participants**    We conducted our study on Amazon Mechanical Turk, receiving a total of 597 re-

---

[1]For other tasks, if they do not need a sentence pair (e.g., sentiment analysis), our instructions always unambiguously ask for judgment of `sentence1`.

[2]W&P further differentiate moderately misleading from extremely misleading instructions. However, for our purposes, we collapse this distinction, except where discussed in §A.2.

[3]See Appendix E for a detailed discussion and analysis of control conditions which we take to be convincing evidence

Christopher Reeve, an actor and director who became an inspiration worldwide after being paralyzed in a horse riding accident, died Sunday of heart failure.

Question: Does this imply that, Christopher Reeve had an accident.

☐ Yes ☐ No

{Inst. w.r.t. NLI: Instructive
Suface instruction: NLI
Surface label: Yes
NLI label: Yes}

(a) Instructive Condition: Baseline where the surface instruction is the same as NLI.

A male rabbit is called a buck and a female rabbit is called a doe, just like deer.

Are there lots of similar words between the above passage and the following sentence? A Female rabbit is called a buck?

☐ Yes ☐ No

{Inst. w.r.t. NLI: Misleading
Surface instruction: Lexical Overlap
Surface label: Yes
NLI label: No}

(b) Misleading Condition: Surface task label always differs from the NLI label.

Soprano's Square: Milan, Italy, home of the famed La Scala opera house, honored soprano Maria Callas on Wednesday when it renamed a new square after the diva.

Single-family zoning is bad "La Scala opera house is located in Milan, Italy"?

☐ Yes ☐ No

{Inst. w.r.t. NLI: Irrelevant
Surface instruction: None
Surface label: N/A
NLI label: Yes}

(c) Irrelevant/Null Cond.: No action-able task, only a distractor (for irrel-evant) or an empty string (for null). Above shows the irrelevant condition.

Figure 1: Main experimental design. Note that under the misleading condition, if a participant interprets the surface instruction as lexical overlap, then the gold answer would be "yes". If a participant somehow interprets this instruction as NLI, then the gold answer will be "no". Text within the curly bracket are not shown to the participants.

| Category | Prompts | Examples | Total Items |
|---|---|---|---|
| Instructive | 5 | 12 | 60 |
| Misleading | 10 | 5 | 50 |
| Irrelevant | 5 | 12 | 60 |
| Null | 2 | 12 | 24 |
| Total | 22 | 46 | 194 |

Table 2: We define an *item* as a unique 3-tuple `<sentence1, instruction, sentence2>` where each sentence pair is an *example* manually selected from a dataset.

sponses over a three-day span. Participants were paid a base rate of $0.50 with an additional $0.10 per correct answer as an incentive. See Appendix E for details on how we qualify participants.

**Models** To compare human performance with that of models, we use instruction-tuned models T0++ (11B, Sanh et al., 2022) and Flan-T5 (11B, Chung et al., 2022), as well as GPT-3.5 (`gpt-3.5-turbo`) and GPT-4 (June 2023 version, OpenAI 2023). T0++ and Flan-T5 models are extensively fine-tuned to follow NLP instructions, with the key difference that T0 has NLI as a held-out task while Flan-T5 does not. All models are given test items identical to those from the human experiments. (See §E.4 for additional details).

## 3 Results

Figure 2 show the zero-shot performance of humans compared to that of instruction-tuned

models. The overall performance of T0++, Flan-T5 and GPTs by themselves is consistent with the model performance reported by W&P. But, with the exception of GPT-4, models show very different patterns from that of humans when given misleading prompts. As expected, when humans are explicitly asked to do a task other than NLI (e.g., sentiment analysis, grammaticality), they tend to do the specified task (leading to low accuracy when measured against the NLI task). In contrast, models often appear to behave as though they have been instructed to do NLI even though they are instructed to do some other surface tasks (leading to high accuracy when measured against the NLI task). For example, when given a misleading instruction (e.g., paraphrasing identification as opposed to NLI) `{sentence1}` `Does that have the same meaning` `as "{sentence2}"?`, humans do indeed perform the paraphrasing task and thus receive a score of 0 on NLI, whereas models tend towards performing NLI, with T0++/GPT-4 receiving an NLI score of 1 and Flan-T5/GPT-3.5 a score of 0.6. (Full results on all prompts in §G.1.) This pattern confirms W&P's assumption A1 that models perform 'too well' on misleading prompts, i.e., better than humans would under similar conditions.

However, when we consider assumption A2 (instructive > irrelevant), we see a different story. When given irrelevant instructions, we see that all models and humans exhibit similar patterns. In fact, humans show far less variance than models in performing the NLI task when

---

that our participants are sufficiently competent.

[4] We add a line break after the premise text for better human readability, and we edit all prompts to be clearer by making use of the line break (e.g. "the above passage").

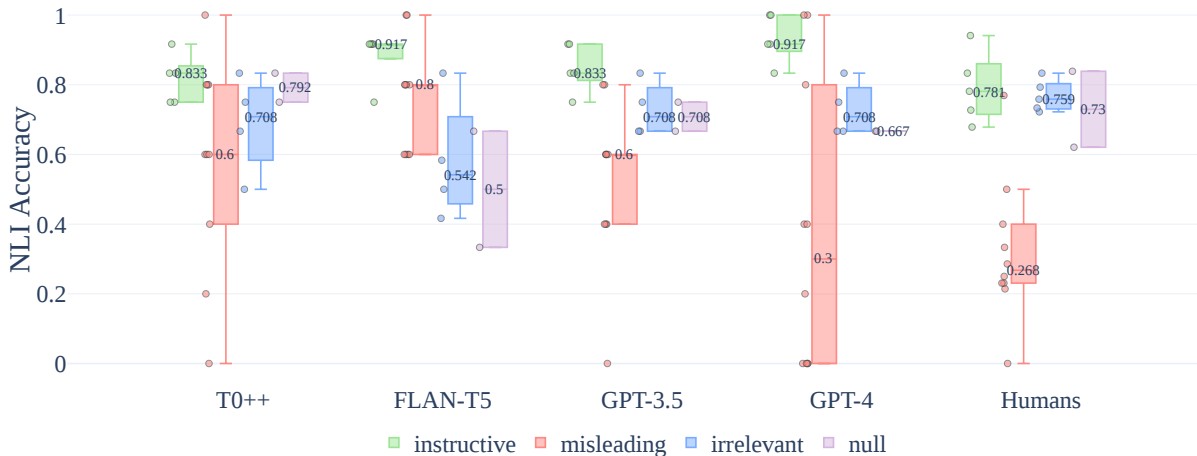

Figure 2: Zero-shot accuracy of human annotators vs. instruction-tuned models. In the misleading condition, lower NLI accuracy is preferred as it means higher accuracy in following the instructions to perform the surface tasks. Medians are displayed in the bars. Each scatter point represents the accuracy on a instruction within the semantic category. For models, this is calculated as the mean accuracy over all the test items constructed from the examples for the instruction (i.e., 5 for each misleading instruction, 12 for each null, irrelevant and instructive instruction); for humans this is the mean accuracy over all participants who received items with that instruction.

the instructions are irrelevant, suggesting that humans are more likely to perform NLI as some kind of "default" task absent of useful instructions. For example, given the irrelevant instruction `{sentence1} Inflections are annoying and thank god that Middle English got rid of most of them. {sentence2}` Humans, T0++, Flan-T5, GPT-3.5 and GPT-4 all score similarly at 0.79, 0.83, 0.83, 0.67 and 0.75 respectively (§G.1).

## 4 Discussion

Our findings show that, in a zero-shot setting, humans appear largely faithful to prompt instructions. They perform well given instructive prompts and poorly given misleading prompts (as expected). However, we observe that T0++, FLAN-T5, and (to a lesser extent) GPT-3.5 are inclined to perform NLI in a zero-shot setting regardless of what is in fact being instructed. GPT-4, however, does seem to exhibit a more human-like pattern in following the misleading instructions (albeit with high variance). It is possible that a combination of pretraining FLOPs, fine-tuning data quality, and RLHF may have bridged some discrepancies between GPT and human behaviors, whereas smaller and supervised fine-tuning-only models fail to do so, but it is impossible to conclude given that little is known about GPT-3.5/4's technical details.

When no useful instructions are provided in the irrelevant prompt setting, our results show that humans still tend to perform the NLI task surprisingly well. Contrary to W&P's criticism, models' ten-

dency to do the same with irrelevant prompts is likely more a feature than a bug (cf. Merrill et al., 2022).

Such idiosyncrasies in human behavior are often difficult to anticipate and even more difficult to codify in a way that lends itself well to benchmarks. Thus, we echo recent work (Pavlick and Callison-Burch, 2016; Ross and Pavlick, 2019; Pavlick and Kwiatkowski, 2019; Dasgupta et al., 2022; Lampinen, 2022) in emphasizing the difficulty of evaluating models when "human-likeness" is the intended gold standard. As NLP models become increasingly advanced, and evaluation tasks increasingly complex, we are likely to face increasing challenges in determining whether models' behavior is "aligned" with that of humans. This study contributes to the line of work which underscores the importance of empirically measuring, rather than presupposing, human behavior in such settings, as humans in practice routinely evade basic intuitions. Appendix B further discusses how to design a fairer comparison between humans and models.

## 5 Conclusion

In this work, we measure human behaviors when given misleading and irrelevant instructions for various NLP tasks. We show that our prior work used oversimplified assumptions of human behavior to evaluate NLP models. Our results underscore the need to empirically validate assumptions about human behavior, which is often more complex in reality than our intuitions would lead us to believe.

## 6 Limitations and Future Work

Our main experiment investigates only the zero-shot setting in humans. We do run a pilot experiment with the intention to compare model and human behavior in a few-shot setting, reported in Appendix A. While the pilot already yielded interesting results, we leave a full experiment on the few-shot setting to future work.

Our experimental design also only uses the Natural Language Inference (NLI) task as the reference task, which we acknowledge may be a task that is perhaps more intuitive as a "default" task than other common NLP tasks (e.g., sentiment analysis, paraphrase). While future work should consider repeating this analysis for other tasks, NLI has several advantages for our purposes. First, NLI is one of the only tasks explicitly held out entirely from the instruction-tuned T0 models, allowing for the best evaluation of their few-shot performance. Moreover, the intuitiveness of NLI works in our favor for the claims we make—it would be ostensibly most challenging for humans to override a bias towards this task to instead follow task instructions. Then, the fact humans do this even under the NLI task setting is the strongest evidence of instruction-following behavior compared to any other task. We acknowledge that there is future work to verify that the human behavior observed for instruction-following extends to other tasks.

## 7 Ethics Statement

We acknowledge that calling for more rigorous human benchmarking only exacerbates NLP field's needs for human annotators, where it has been demonstrated that NLP crowdsourcing may potentially expose workers to harm, as described in Shmueli et al. (2021). For our study, our Institutional Review Board (IRB) reviewed our experimental design and determined that its primary aim is to study computational language models and thus does not meet the federal definition of human subjects research. Future studies should similarly submit their studies, if involving human benchmarking, for review by their institutions' IRBs.

## 8 Acknowledgments

We thank Tal Lizen, Andrew Lampinen, Swaroop Mishra, Apoorv Agarwal, Michael Lepori, Louis Castricato, Samuel Musker, Aaron Traylor, and Philip LaDuca for discussions and comments on this work. We thank Brendan Ho who wrote a lot of feedback for the free-form response questions. Special thanks to Roman Feiman for discussions on the design of the human experiments.

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

# Appendix

## Table of Contents

## A   Pilot Few-Shot Experiment

Our main experiment above focuses only on the zero-shot setting. However, model behavior (and likely human behavior as well) can look very different when given even a single labeled example. Thus, we additionally conduct a pilot experiment to investigate human behaviors under a few-shot learning setup. We consider two conditions. The first, *few-shot with labels*, simulates the usual few-shot learning setting: the participant is given the gold label as feedback after answering each question. The second condition, *few-shot without labels*, is inspired by Min et al. (2022)'s study in few-shot learning with corrupted labels. In this condition, we do not give participants any label feedback. However, they still have the opportunity to refine their understanding of the instruction via sequential encounters with different unlabeled examples.

### A.1   Method

We show each subject 32 items sequentially, one item per page, for which subjects must respond "Yes" or "No" to each item. Each subject is assigned an instruction as condition and the instruction is the same across all examples they receive.

In the *few-shot with labels* condition, subjects are shown whether their answer was correct for that item. Here, correctness is always based on NLI (i.e., whether sentence 1 entails sentence 2) regardless of the surface instructions. Unlike the zero-shot study, there is no special design in the example sentence pairs in relation to the instruction, only that the sample is balanced in the number of entailments and non-entailments. That is, a 50% score indicates random performance on the implicit NLI task.

In the *few-shot without labels* condition, subjects do not receive any feedback throughout the 32 items. This setup is inspired by Min et al. (2022), who show that models can surprisingly well learn from corrupted labels. We conjecture that unsupervised exposure to examples may help humans pragmatically infer what researchers want out of them and what the underlying task is.

**Post-Experiment Survey**   After participants complete the 32 items, we ask for paragraph-length responses to two questions: (1) How did you decide to choose between "Yes" or "No"? (2) What did you think about the instructions we gave?

**Instruction Templates**   We manually choose one instruction from each of W&P's five categories for

this pilot study:

1. {sentence1} Using only the above description and what you know about the world, "{sentence2}" is definitely correct. Yes or no? (Instructive; adapted from MultiNLI's instruction to crowdworkers).

2. {sentence1} Can the above be paraphrased as: "{sentence2}" (Moderately misleading.)

3. {sentence1} Is the next sentence grammatically correct? {sentence2} (Extremely misleading.)

4. {sentence1} Single-family zoning is bad for American cities {sentence2} (Irrelevant.)

5. {sentence1} {sentence2} (Null; empty instruction baseline.)

**Participants**   Subjects were undergraduate students ($n = 8$ for the with labels condition, $n = 5$ for the without labels condition). Subjects were asked to finish all 32 items in a single session within an hour. Subjects were paid a base compensation of $15, with a $0.25 bonus for every correct answer as an incentive. As it is expensive to have a participant complete 32 items continuously in a single session (in order to mimic models' few-shot training), and because the trend was sufficiently clear from our pilot experiment, we did not proceed with a larger pool of participants.

## A.2   Results

Figure 3 shows the cumulative scores of subjects across 32 items are shown for both few-shot *with labels* and *without labels* conditions, where each line is one subject. Solid lines represent the performance of subjects with labels, and dotted lines represent the performance of subjects without labels. The subject who received the instructive prompt achieved a perfect score, and their performance is presented as a green reference $y = x$ line against all participants who received other prompts.

**Irrelevant Instructions**   Participants who received the irrelevant instruction perform practically identically with or without label feedback. In both conditions subjects with the irrelevant prompt also performed almost just as well as the subject who received the instructive prompt. Like in the zero-shot study, this pattern provides evidence against W&P's assumption A2 (instructive > irrelevant).

The post-experiment survey also confirms that participants were able to figure out that the prompt was simply irrelevant (Appendix K).

**Paraphrasing Inst. (Misleading)**   When a participant is given a paraphrasing instruction without label feedback, they were successfully misled to perform paraphrasing identification, thus scoring much lower on NLI (dashed yellow line in Figure 3). But when a participant is given a paraphrasing instruction with NLI labels as feedback, they quickly adapted their interpretation of the instruction in order to fit the labels (solid yellow line in Figure 3). As one participant wrote in the post-experiment survey:

> In the first few questions, my strategy is to read through the entire paragraph or sentence and then decide whether the paraphrased sentence makes sense or not. However, then I started to look at the paraphrased sentence first and decide whether it is correct or wrong based on the given piece of text.

That is, this participant completely recovered the NLI instruction even though they were given a paraphrasing instruction. They even observed the unidirectional entailment nature of NLI (i.e., sentence2 does not need to entail sentence1) vs. the bidirectional entailment nature of paraphrasing (i.e., the two sentences must express approximately the same meaning):

> Initially, I also considered whether the paraphrased sentence captured all the major details or not, but the quiz later shows that comprehensiveness is not a factor.

**Grammaticality Inst. (Misleading)**   While paraphrasing is a task related to NLI, grammatical acceptability is a much more misleading instruction since it has nothing to do with NLI. Here, we see the with and without label results nearly reversed: when given NLI label feedback, the grammatical acceptability instruction appears to be so incompatible with the labels that one participant was confused and unable to adapt their interpretation to just one task, and ultimately scored much lower on NLI (lower solid red line in Figure 3). In the post-experiment survey, they wrote

> I basically went on a gut reaction on what was correct. On some weird instances, I

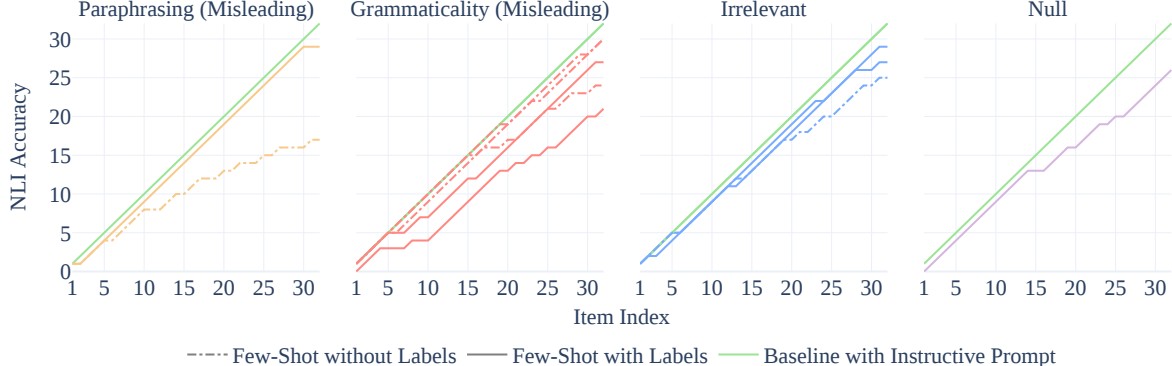

Figure 3: Cumulative total scores between the *few-shot with labels* and *few-shot without labels* experiments, within categories. The y-axis plots the sum of correct answers out of $x$ items answered so far. The sequence of 32 prompts were the same across both experiments for all prompt categories. Each line is one subject.

thought hard about good grammar. Oftentimes, I looked at the content of the sentence. If it was factually correct, I guessed yes.

However, another participant did eventually figure out the underlying NLI task:

I feel like the question should be changed because it seems like the question is actually, "Is this statement true based on the context given in the paragraph."

When no label feedback is giventhe results are more surprising and nuanced. Because all of our examples are handpicked and grammatical, participants found themselves answering "yes" to many examples and starting to question the instruction itself:

I looked at whether the sentence was accurate to the information given in the text, and also if the sentence itself had correct grammatical structure. It was a little difficult because some of the sentences made inferences that weren't explicit in the given text, so I wasn't sure if that was a grammatical error or not.

They then started to incorporated the *semantic* well-formedness into their interpretation of grammatical acceptability:

Usually, I think of something as being grammatically correct when the sentence has correct grammatical structure, including punctuation and capitalization. Since most of the sentences seemed to fit this,

I thought that maybe grammar also encompasses the validity of the statement based on the text, so I chose my answers based on that.

Additional survey responses are included in Appendix K. Overall, human behavior in the few-shot cannot be readily summarized as a single inequality as attempted in A1 - 3. Rather, different participants can respond to the same instructions in different ways; some interpret the instructions strictly and literally, while others adopt a more relaxed pragmatic interpretation, and still others refine their interpretations over time.

## B Reconciling Few-Shot and Zero-Shot Evidence

As we find instruction-tuned models seem to largely match human behaviors in the few-shot setting, while falling short of human in the zero-shot setting, which setting should we weigh more for evaluating models' understanding of prompts as instructions? Concurrent work by Lampinen (2022) notes that zero-shot could be an inherently problematic way to study the full competence of LMs: From a model's perspective, it has just finished imitating (for example, in T5's case) a trillion tokens of highly heterogeneous content and linguistic styles with communicative intents far from answering academic evaluations. In a zero-shot setting, it could be unclear to the model "what is the intended continuation; the model might be likely to produce a blank line for someone to fill in the answer, or jokes mocking the question, or just arbitrary rambling" (Lampinen, 2022, p. 7); all of the above may be valid continuations for its language modeling

pretraining objective. We partially address this issue by only using instruction-tuned models, which are trained on a large mixture of traditional NLP datasets and thus primed to directly answer the question.

On the human side, in order to make a fair comparison, we carefully design our experiments to make sure the human responses are "as zero-shot as possible" by (1) having one participant answer only one test question and (2) having all qualification questions come after the test question so as to not bias the participants. This is a highly controlled, perhaps even contrived, condition that does not reflect well on how humans learn from instructions in the real world, or even in most other cognitive science experiments where participants are often given familiarization trials prior to the test trials.

Therefore, we agree with Lampinen (2022) that, for future studies, the few-shot setting is likely a more productive way to probe a model's true competence, even though it may be scientifically less controlled in other respects, since now effects of the few-shot examples must be considered, which could also be counter-intuitive as shown in Appendix A and Min et al. (2022).

However, zero-shot evidence should not be ignored either, especially considering that there is a consistent collection of work showing language models being insensitive to instructions on tasks in addition to NLI and on models of various sizes and fine-tuning strategies.

## C  Related Work

**Zero-Shot Instructions**  In addition to W&P, many papers find similar results that models perform well with semantically incoherent instructions on a variety of tasks. Prasad et al. (2022) find that semantically incoherent prompts work well for InstructGPT over 12 QA and coreference datasets in Natural Instructions Mishra et al. (2021), Khashabi et al. (2021) find this to be true for GPT-2 on 5 sentiment analysis and topic classification datasets in Natural Instructions. Jang et al. (2022) show OPT and InstructGPT are unable to follow negated instructions over 9 QA and sentence completion datasets, performing well below human baseline of 13-year-olds.

Note that none of the above papers, including W&P, claim that pathological prompts would perform just as well for all tasks. Indeed, Kojima et al. (2022) show various irrelevant and misleading baselines perform poorly on an arithmetic dataset (Roy and Roth, 2015). Instead, W&P claim that the existence of high-performing pathological prompts for a large number tasks show that they use prompts in an un-human-like way, while the existence of bad-performing pathological prompts is orthogonal to this line of argument.

**Few-Shot Exemplars and Explanations**  Unlike zero-shot, the few-shot setting has more conflicting evidence in the literature on whether models perform just as well with pathological prompts. Min et al. (2022) first showed that the correctness of the few-shot labels are not required, concluding that prompts are largely helping models to adapt to the domain and format of the input text as well as the space of possible labels.

As few-shot exemplars are now commonly accompanied with intermediate computation, explanations, or chain-of-thoughts (Wei et al., 2022; Nye et al., 2021; Zhou et al., 2022; Lampinen et al., 2022), Madaan and Yazdanbakhsh (2022) agree with Min et al. (2022) and find that various few-shot chain-of-thought prompts—with corrupted symbols but retaining the overall task format—have performance comparable to non-corrupted baseline, thus arguing that "CoT helps a language model in imitating the prompt and generating the right tokens for the task—and is conceivably less related to their reasoning abilities." Similarly, Ye and Durrett (2022) find that explanations improve performance modestly for OPT and GPT-3 but improve substantially for text-davinci-002 on 3 QA and NLI datasets. However, the model-generated explanations themselves are often inconsistent or factually incorrect, concluding that "model internal 'reasoning' does not always align with explanations that it generates."

With extensive statistical controls on a diverse range of BIG-Bench tasks (Srivastava et al., 2022), Lampinen et al. (2022) find that LMs' success with post-answer explanations do outperform baselines such as same-length word-scrambled explanations, domain-relevant but non-explanatory statements, and correct explanations misaligned with wrong few-shot examples. Notably, although they find a positive result with the effect of explanations, they also find that models are much more insensitive to the effect of instructions.

## D Author Contributions

Albert Webson led the project, co-designed the experiments, implemented the model evaluation code, and led the paper writing.

Alyssa Marie Loo co-designed the experiments, manually curated all prompts and examples, evaluated all models, produced all analyses, and co-authored the paper.

Qinan Yu co-designed the experiments, conducted all human experiments, and co-authored the paper.

Ellie Pavlick advised the project and edited the paper.

## E Additional Details of the Main Zero-Shot Experiment

### E.1 Controls

We collect additional data to test the robustness of our result on different subgroups of participants selected under multiple conditions. After the test condition, participants are asked to complete two additional control conditions, *General (Surface Tasks) Control* and *NLI Control*, that provide us such selection criteria for post-hoc analysis in Appendix F. These controls test whether the same trend still holds under more restrictions such as receiving perfect scores on both controls as additional analysis. However, note that our main results do not exclude any participants using these controls as it may bias the results to only consider participants with specific behavior patterns.

The two controls are four items presented after the first item from the test condition, adding up to the five items that each subject is presented in the study. The control items are shown one by one in the following sequence: two items in the *General Control*, and then two items in the *NLI Control*. As with the test condition item, subjects are asked to answer "Yes" or "No" in response to each control item. The performance of our subject population on the controls is shown in Appendix I.

**General Control**  The General Control verifies if participants can perform the misleading tasks. In this control, subjects are scored on whether they perform the surface task correctly. If subjects were already given a misleading prompt in their test condition, in this control they are shown the same prompt with two new examples. Otherwise, subjects are randomly assigned two items with a misleading prompt.

We curate examples from a range of datasets such that the misleading task and NLI task have opposing answers, and also to be converse to the test condition. That is, if the misleading task answers for a prompt is "Yes" in the test condition, it would be "No" in the General Control. See Appendix H for how examples were selected.

**NLI Control**  The NLI Control verifies if participants can perform the NLI task. In this control, subjects are given two items with an instructive prompt and are scored on performing the NLI task correctly.

### E.2 Qualifications

From the 597 responses, for our main results we exclude data from 93 subjects[5] whose total completion time $t$ for the five-question study is less than one standard deviation from the mean of the sample ($t < 33.01$). Extremely low response times have been shown to be an accurate indicator of careless responding, where data from such "speeders" have been shown to lower data quality (Greszki et al., 2015; Goldammer et al., 2020). Past studies have typically defined floor cutoffs based on the distribution of their data, with no singular convention. From our data's response time distribution (Figure 4), there is a clear bimodal distribution between "speeder" response times and typical response times; a floor of $t = 33.01$ sufficiently excludes the "speeder" distribution to leave a sizeable sample of $n = 504$.

We also ran a separate pilot study in order to determine whether previous exposure to other NLP studies on Mechanical Turk would bias subjects to perform the underlying NLI task. The results were inconclusive and thus we decided not to exclude participants who had participated in NLP studies previously. See Appendix J.

### E.3 Instructions to Workers

Workers will see the following instructions before they begin the study.

> You will be given 5 short paragraphs on separate pages, each containing a yes-no question. You will be paid at least $0.5 in addition to an extra $0.1 for each correct answer.
>
> If you have a high number of correct answers, you will be qualified to participate in

---

[5] Even if a subject's data was excluded, they were still compensated.

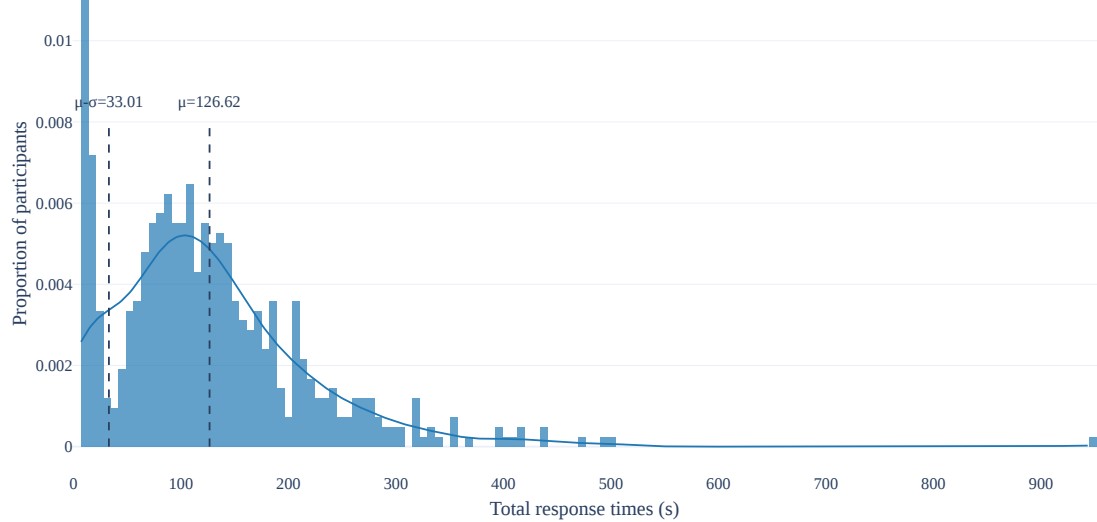

Figure 4: Histogram of subjects' total time taken (in seconds) to complete the zero-shot study ($n = 597, \mu = 126.62, \sigma = 93.62$). The left dotted line is the floor cutoff ($t = 33.01$): the 93 participants that had a total response time lower than this cutoff were excluded from the main paper's analysis.

> our full, \$15 per hour study. You will not be graded based on how fast you finish.
>
> You have 10 minutes to answer all 5 questions, which should be plenty of time, but please complete our study in one continuous session and do NOT navigate to other tabs or tasks, as we are measuring the time it takes you to respond to each question.
>
> You can only work on this HIT once. Multiple submissions will be prevented.

### E.4 Evaluation Protocol Details

Following Sanh et al. (2022) and Brown et al. (2020), we evaluate T0++ and Flan-T5 by rank classification. We evaluate GPT-3.5 and GPT-4 by string match, as it is no longer possible to get per-token token log-probabilities from OpenAI APIs. Fortunately, GPTs are able to follow instructions that return "Yes" or "No" as its first token, so we use greedy decoding (temperature = 0) which is equivalent to single-token rank classification.

Each item is input into the GPTs as a content message in the `user` role, with the `system` prompt "You will be given a short paragraph with a yes-no question. You strictly must answer only 'Yes' or 'No' to the paragraph.". No other instructions or examples are given.

## F Additional Results for Zero-Shot Experiment

We present post-hoc analysis of our zero-shot experiment data using different subgroups of participants. For all figures in this section, medians are indicated in the bars. Each scatterpoint represents accuracy on a prompt within the semantic category, calculated as the aggregated accuracy over multiple annotators whose test item was constructed by the prompt and one of the prompt's five possible examples.

### F.1 Using Control Scores as Qualification

In Figure 5, Figure 6, Figure 7 and Figure 8 we present results using the controls as exclusion criteria.

Constraining data to subjects that score perfectly on the General Control (Figure 6) decreases human performance with misleading prompts on the NLI task in the test condition dramatically—that is, these subjects perform the non-NLI task extremely well right from the first item, given a prompt that instructs a misleading task. This sample likely selects for subjects that interpret instructions most strictly.

In contrast, results from constraining data to subjects that score perfectly on the NLI Control (Figure 5) remain highly similar to our main results. This supports that it is not incompetence with the NLI task that causes humans to score poorly in the test condition if given a misleading prompt, but that they are indeed following the given instructions to perform the misleading task.

### F.2 Without Response Time Qualification

In Figure 9, Figure 10, Figure 11, Figure 12 and Figure 13 we present results without using response time qualification. Removing the response time qualification has the effect of introducing noise into the data: the overall variance between different prompt categories reduces as all categories tend towards 0.50. However, the top-line trends we argue in the main paper about human behavior given these different prompts remain observable (instructive > misleading-moderate ; instructive > misleading-extreme ; instructive ≈ irrelevant).

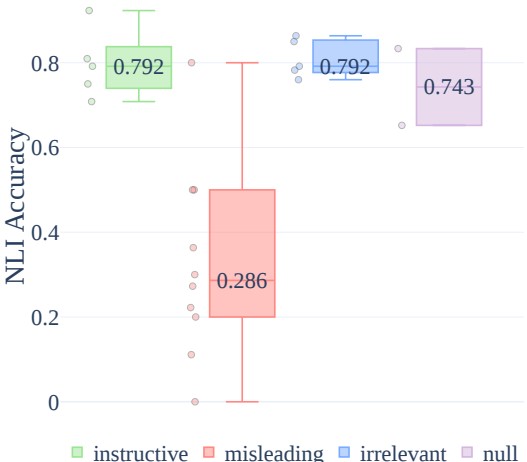

Figure 5: Zero-shot accuracy data of human annotators with perfect NLI Control scores and time above floor cutoff. ($n = 384$)

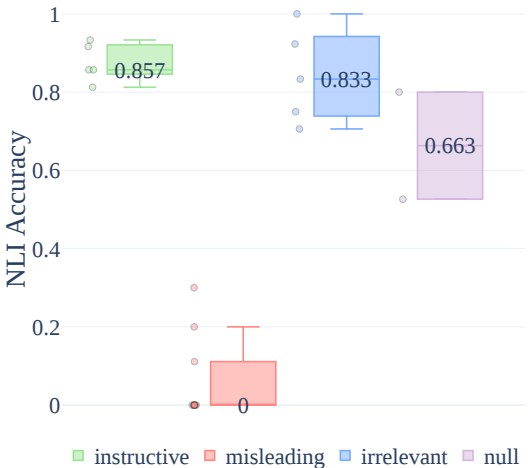

Figure 6: Zero-shot accuracy data of human annotators with perfect General Control scores and time above floor cutoff. ($n = 238$).

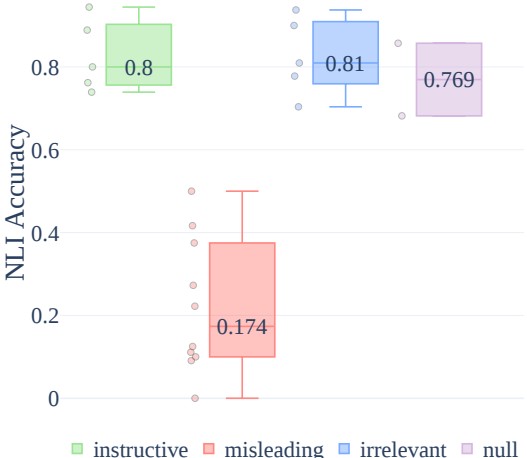

Figure 7: Zero-shot accuracy data of human annotators with a total score of at least 3 out of 4 Control items and time above floor cutoff. ($n = 340$).

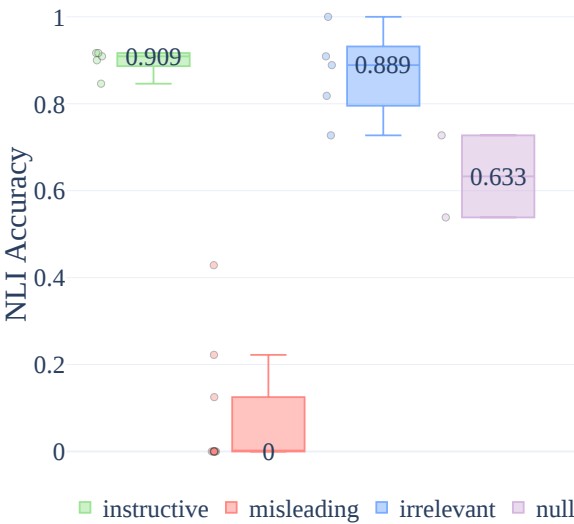

Figure 8: Zero-shot accuracy data of human annotators with all perfect NLI and General Control scores and time above floor cutoff. ($n = 186$).

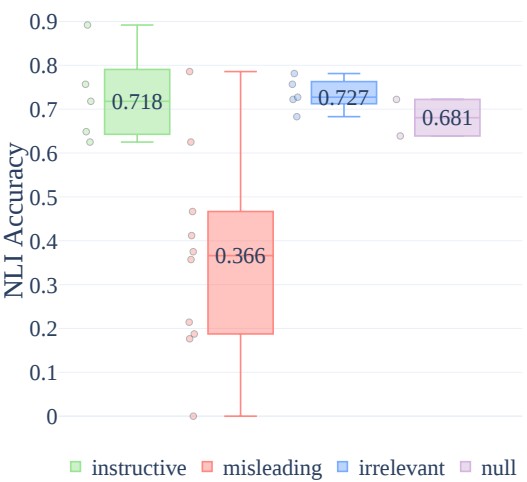

Figure 9: Zero-shot accuracy data of human annotators with any response time ($n = 597$) (i.e., all data).

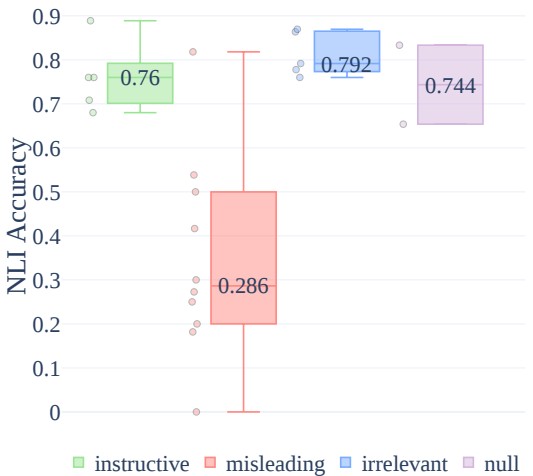

Figure 10: Zero-shot accuracy data of human annotators with perfect NLI Control scores, with any response time. ($n = 409$).

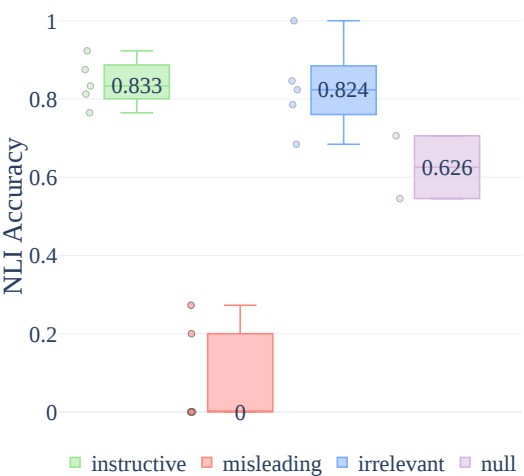

Figure 11: Zero-shot accuracy data of human annotators with perfect General Control scores, with any response time. ($n = 273$).

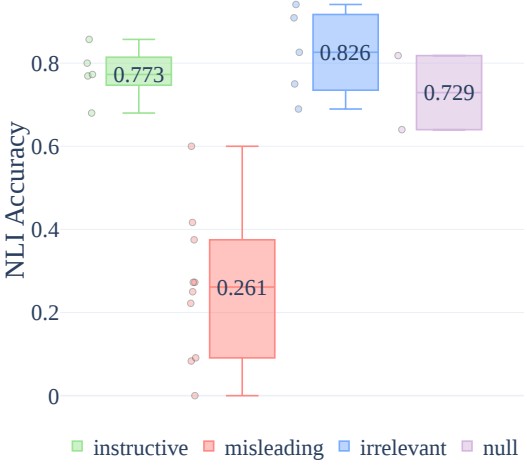

Figure 12: Zero-shot accuracy data of human annotators with a total score of at least 3 out of 4 Control items, with any response time. ($n = 377$).

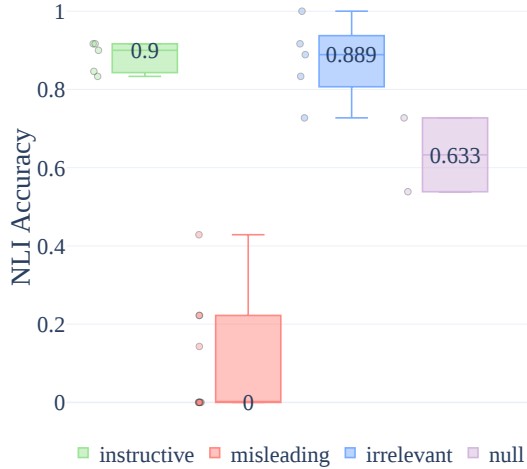

Figure 13: Zero-shot accuracy data of human annotators with all perfect NLI and General Control scores, with any response time. ($n = 192$).

# G  Additional Figures and Data for Zero-Shot Experiment

## G.1  Accuracies Per Instruction

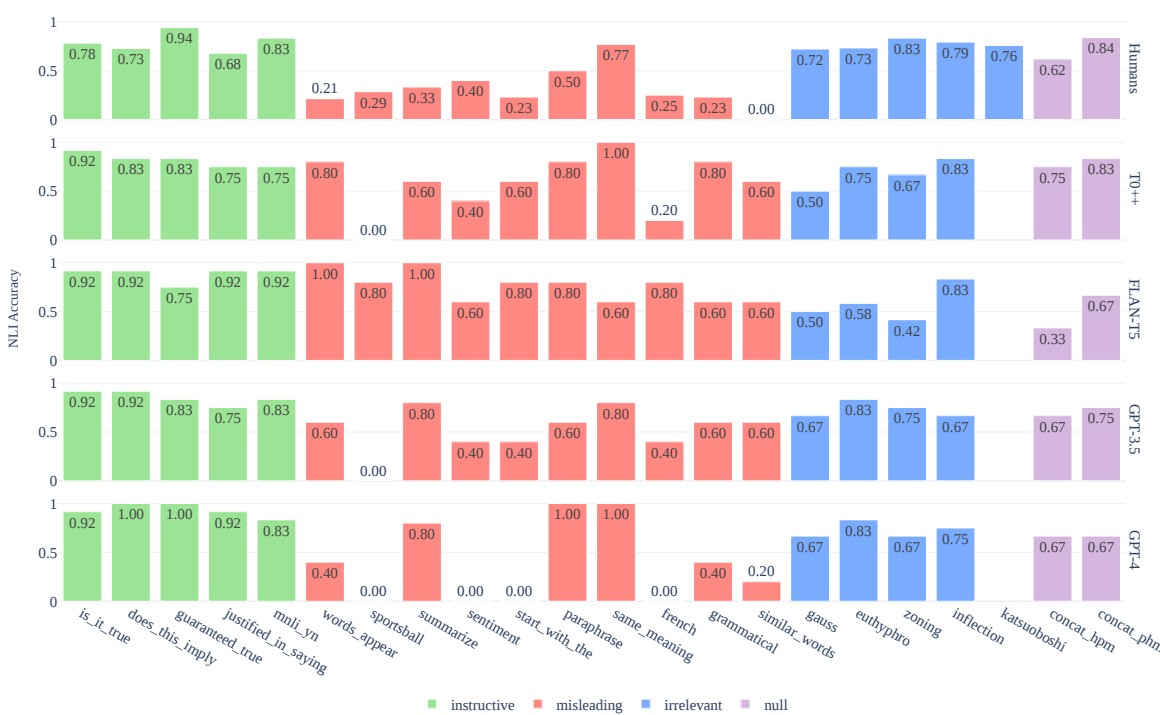

Figure 14: Zero-shot NLI accuracy of human annotators vs. models, by specific instruction (see §H.1 for instruction templates). Bars are colored by instruction category.

# H Instructions and Examples

## H.1 All Instructions

| category | name | instruction |
|---|---|---|
| instructive | MNLI-YN | {sentence1} Using only the above description and what you know about the world, "{sentence2}" is definitely correct. Yes, no or unclear? |
| instructive | justified-in-saying | {sentence1} Are we justified in saying that "{sentence2}"? |
| instructive | is-it-true | {sentence1} Based on the previous passage, is it true that "{sentence2}"? |
| instructive | guaranteed-true | {sentence1} Then, is it guaranteed true that "{sentence2}"? |
| instructive | does-this-imply | {sentence1} Question: Does this imply that "{sentence2}"? |
| misleading | words-appear | {sentence1} Do most of the words in the following sentence appear in the above paragraph? {sentence2} |
| misleading | similar-words | {sentence1} Are there lots of similar words between the above passage and the following sentence "{sentence2}"? |
| misleading | same-meaning | {sentence1} Does the above have the same meaning as "{sentence2}"? |
| misleading | paraphrase | {sentence1} Can the above be paraphrased as: "{sentence2}"? |
| misleading | summarize | {sentence1} Can the above be summarized as "{sentence2}"? |
| misleading | start-with-the | {sentence1} Does the previous paragraph start with "the"? {sentence2} |
| misleading | grammatical | {sentence1} Is the next sentence grammatically correct? {sentence2} |
| misleading | sentimment | {sentence1} Is the above paragraph a positive review? {sentence2} |
| misleading | sportsball | {sentence1} Is the above paragraph a piece of sports news? {sentence2} |
| misleading | french | {sentence1} Is the above text French? {sentence2} |
| irrelevant | zoning | {sentence1} Single-family zoning is bad for American cities. "{sentence2}"? |
| irrelevant | inflection | {sentence1} Inflections are annoying and thank god that Middle English got rid of most of them. "{sentence2}"? |
| irrelevant | gauss | {sentence1} When Bolyai sent Gauss his discovery of non-Euclidean geometry, Gauss replied that he arrived at the same results 30 years ago. "{sentence2}"? |
| irrelevant | katsuoboshi | {sentence1} If bonito flakes boil more than a few seconds, the stock becomes too strong? "{sentence2}"? |
| irrelevant | euthyphro | {sentence1} Is the pious loved by the gods because it is pious? Or is it pious because it is loved by the gods? "{sentence2}"? |
| null | concat-phm | {sentence1} {sentence2} |
| null | concat-hpm | {sentence2}{sentence1} |

Table 3: All prompts used in the main text of the paper. All templates use "Yes"/"No" as target words for the entailment and non-entailment classes, respectively.

## H.2 Example Sources for Test Condition

| category | name | dataset | remarks |
|---|---|---|---|
| instructive
instructive
instructive
instructive
instructive | MNLI-YN
justified-in-saying
is-it-true
guaranteed-true
does-this-imply | RTE, MNLI | 6 entailment labels, 4 contradiction labels, 2 neutral labels were chosen; the latter two map to "No" answers. |
| misleading
misleading | words-appear
similar-words | RTE | Examples were handpicked such that misleading task would have a "Yes" label while NLI task had "No" label. From RTE labels, 3 contradiction labels and 2 neutral labels were chosen and mapped to the "No" labels. |
| misleading
misleading
misleading | same-meaning
paraphrase
summarize | RTE | Examples were handpicked such that the misleading task would have a "No" label while NLI task had "Yes" label—i.e., through examples where the second sentence was indeed an entailment but only tangential to the main point of the first sentence. |
| misleading | start-with-the | RTE | All examples were such that the premise paragraph did indeed start with 'the' but the hypothesis sentence was not entailed, so the misleading task answer was "Yes" while the NLI task answer was "No". 3 contradiction labels and 2 neutral labels were chosen to map to "No" labels. |
| misleading | grammatical | RTE | Grammatically correct but non-entailing examples from RTE were chosen such that the misleading task answer is "Yes" while the NLI task answer is "No". 3 contradiction labels and 2 neutral labels were chosen to map to "No" labels. |
| misleading | sentiment | Amazon Polarity (Zhang et al., 2015) | Reviews were taken from the Amazon Polarity dataset as premise paragraphs. A hypothesis sentence was manually written based on the review. If the review was positive, the hypothesis sentence was not entailed; if the review was negative the hypothesis sentence was entailed. There were 3 non-entailments and 2 entailments. |
| misleading | sportsball | RTE | RTE examples that had nothing related to sports were chosen, such that the misleading task answer is "No" while the NLI task answer was "Yes". |
| irrelevant
irrelevant
irrelevant
irrelevant
irrelevant | zoning
inflection
gauss
katsuoboshi
euthyphro | RTE, MNLI | 6 entailment labels, 4 contradiction labels, 2 neutral labels were chosen; the latter two map to "No" answers. |
| null
null | concat-phm
concat-hpm | RTE, MNLI | 6 entailment labels, 4 contradiction labels, 2 neutral labels were chosen; the latter two map to "No" answers. |

Table 4: All source datasets used for each prompt for the main text of the paper. All templates use "Yes"/"No" as target words for the entailment and non-entailment classes, respectively. For RTE examples, we collapse the SuperGLUE version's "neutral" and "contradiction" to "non-entailment" such that all of our tasks are binary classification. We balance the distribution of "contradiction" and "neutral" labels within our study's non-entailed items.

## H.3   Example Sources for General Controls

| category | name | dataset | remarks |
|---|---|---|---|
| misleading | start-with-the | RTE | All examples were such that the premise paragraph did indeed start with 'the' but the hypothesis sentence was not entailed, so the misleading task answer was "Yes" while the NLI task answer was "No". 5 contradiction labels were chosen to map to "No" labels. |
| misleading | grammatical | BLiMP (Warstadt et al., 2020) | Grammatically incorrect sentences from BLiMP were used as hypothesis sentences, while grammatically correct premise paragraphs were handwritten for the sentence, such that the misleading task answer was "No" while the NLI task answer was "Yes". |
| misleading | sentiment | Yelp Polarity (Zhang et al., 2015) | Reviews were taken from the Yelp Polarity dataset as premise paragraphs and a hypothesis sentence was manually based on the review. If the review was positive, the hypothesis sentence was not entailed; if the review was negative the hypothesis sentence was entailed. There were 3 non-entailments and 2 entailments. |
| misleading | sportsball | HuffPost (Misra and Grover, 2021; Misra, 2022) | Excerpts were taken from articles in the 'Sports' category of the dataset and non-entailing hypothesis sentences were manually written, such that the misleading task answer was "Yes" while the NLI task answer was "No". |
| misleading | french | XNLI (Conneau et al., 2018) | Non-entailing French XNLI examples were taken such that the misleading task answer was "Yes" while the NLI task answer was "No". |

# I Results on Zero-Shot Experiment Control Conditions

## I.1 NLI Controls

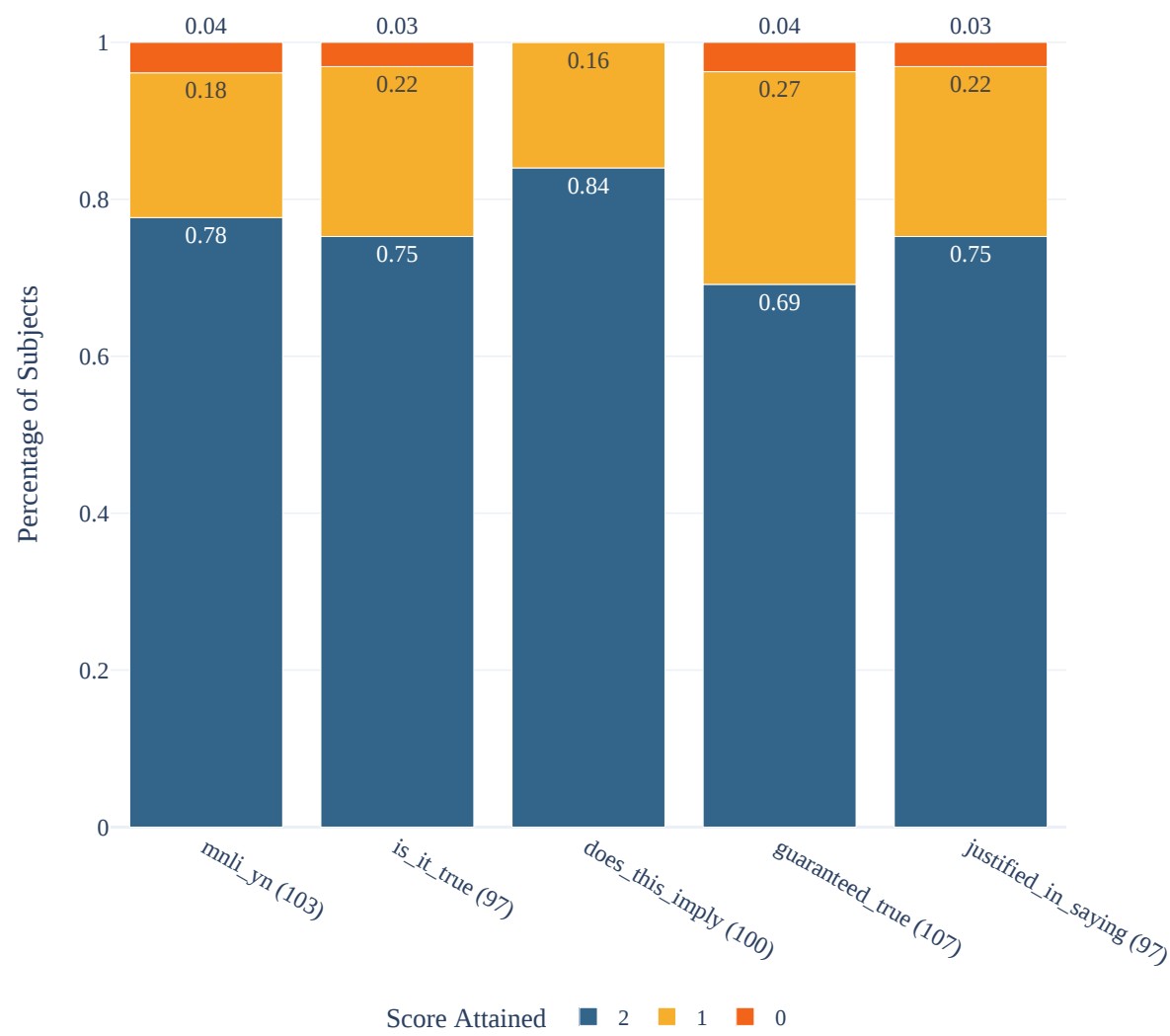

Figure 15: Subjects' scores on the NLI control condition ($n = 504$, only subjects whose completion times were above floor cutoff) Each bar represents the breakdown of percentage of subjects assigned the prompt who scored 0, 1 and 2 out of two NLI control items presented.

## I.2 General (Surface Task) Controls

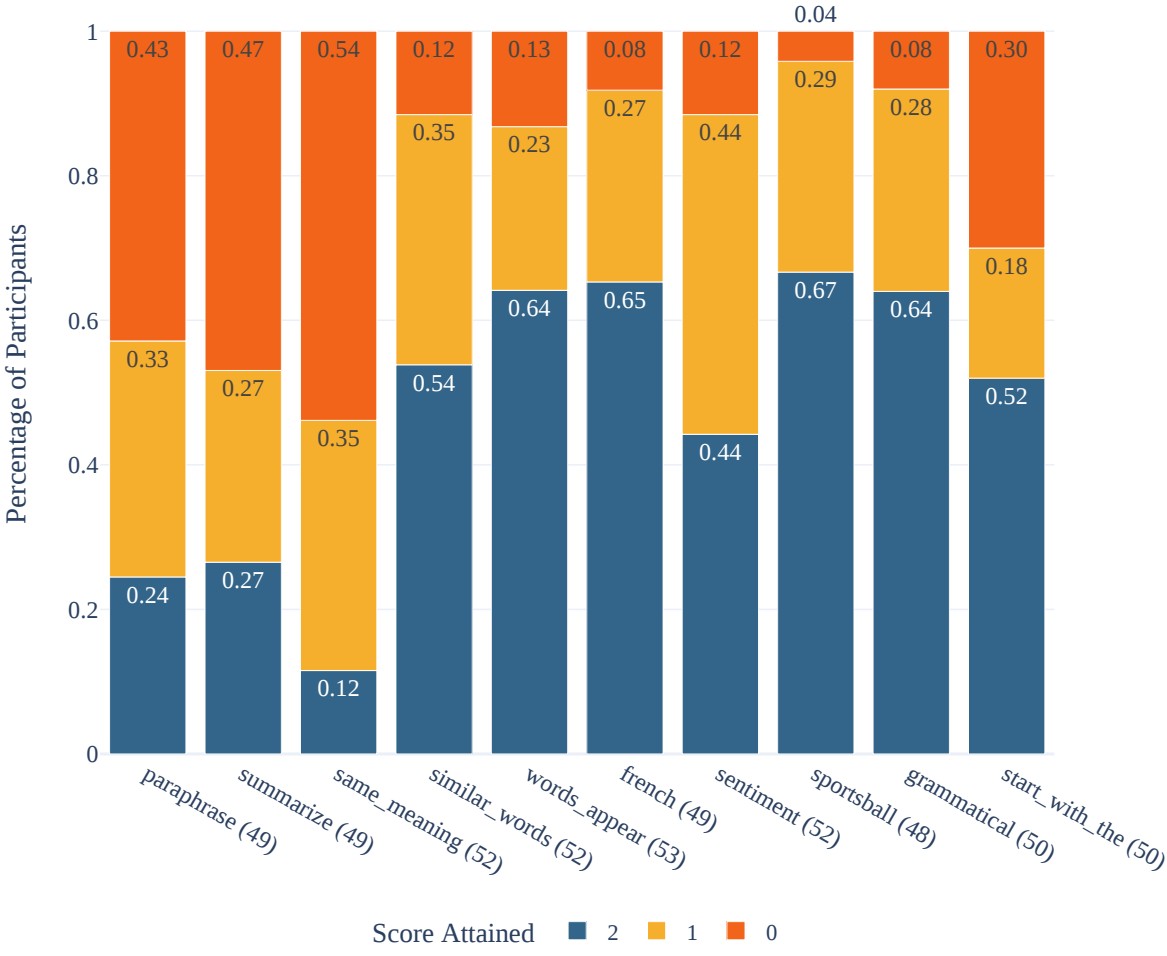

Figure 16: Subjects' scores on the general suface task controls ($n = 504$, only subjects whose completion times were above floor cutoff) Each bar represents the breakdown of percentage of subjects assigned the prompt who scored 0, 1 and 2 out of two general control items presented; subjects were scored on correctly performing the misleading task.

## J Effect of Experience with Prior NLP Studies

We conducted a pilot ($n = 29$) to assess the effect of prior exposure to NLP studies on humans' interpretation of prompt instructions. In this pilot, we select only subjects with no prior experience with NLP studies. All participants had to first screen through a pre-test with the question "How many mTurk tasks have you completed for language research (e.g. Stanford NLP Group, MIT NLP Group, NYU NLP Group, etc.)?". Only participants who selected the option "None" were qualified to continue to take the study. 66 participants took the pre-test and 29 qualified as subjects. We compare these results to an earlier pilot ($n = 67$) that used the same prompts and examples, with no filtering of participants based on previous exposure.

Comparing control condition scores (Figure 17 vs. Figure 18 for NLI Controls' Figure 19 vs. Figure 20 for General Controls'), subjects without prior exposure score higher on both the misleading task and NLI task (recall that in the controls, subjects are scored on performing the surface task as explicitly described by the prompt). Comparing test condition scores (Figure 21 vs. Figure 22), subjects without prior exposure perform better at the NLI task when instructions are instructive and dramatically worse when instructions are misleading, compared to the sample that was not controlled for exposure. These results suggest that subjects without prior exposure appear to follow explicit task instructions more closely than the sample that was not controlled for exposure.

The behavior of subjects without prior exposure to NLP studies is similar to the result when we select subjects that score perfectly on the General Controls (Figure 6)—suggesting that specifying for NLP-study inexperience may select for a sample of humans who follow task instructions more strictly. While we leave a full study that controls for exposure to prior NLP studies to future work, we predict that it will only strengthen the trend for misleading prompts seen in our main results (namely, that humans do poorly on the actual task if given misleading prompts).

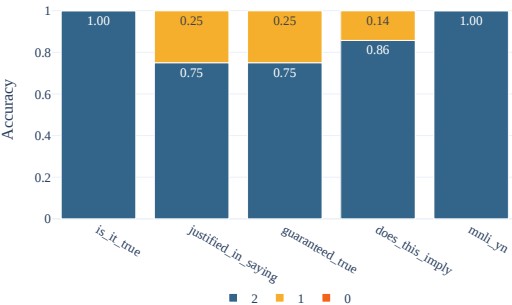

Figure 17: NLI Control scores of subjects ($n = 29$) with no prior NLP experience.

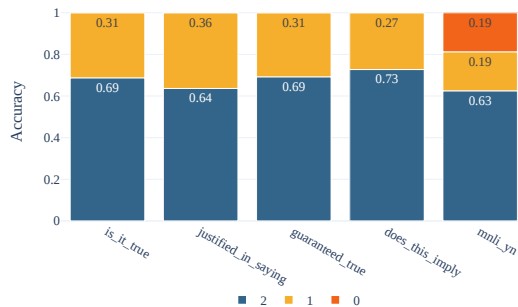

Figure 18: NLI Control scores of sample of subjects ($n = 67$) that were not filtered on prior exposure to NLP tasks.

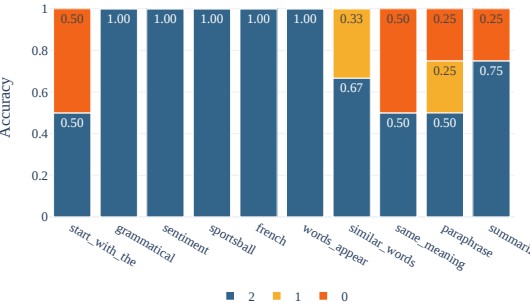

Figure 19: General Control scores of subjects ($n = 29$) with no prior NLP experience.

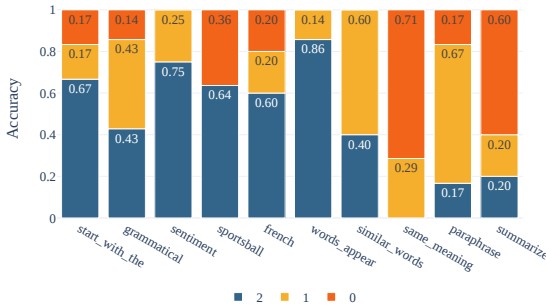

Figure 20: General Control scores of sample of subjects ($n = 67$) that were not filtered on prior exposure to NLP tasks.

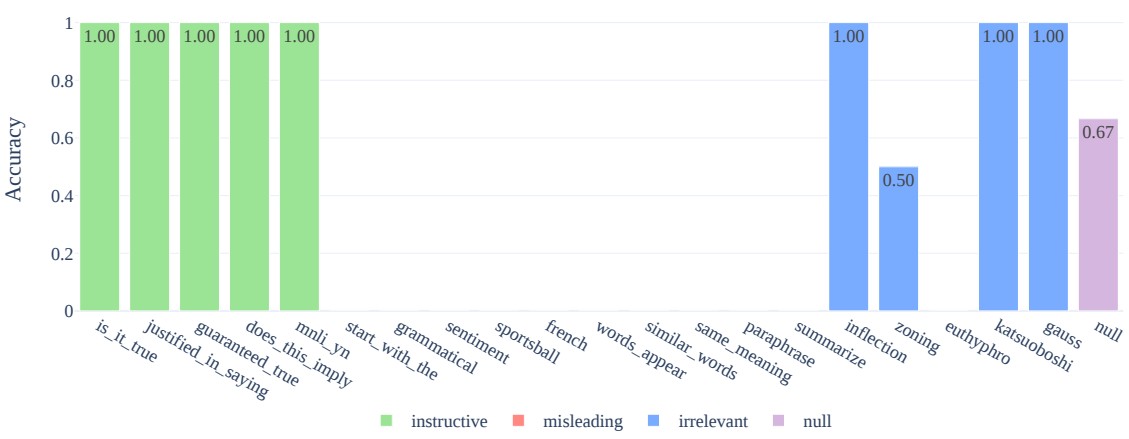

Figure 21: Per-instruction accuracy on the test condition item of subjects with no prior NLP experience ($n = 29$).

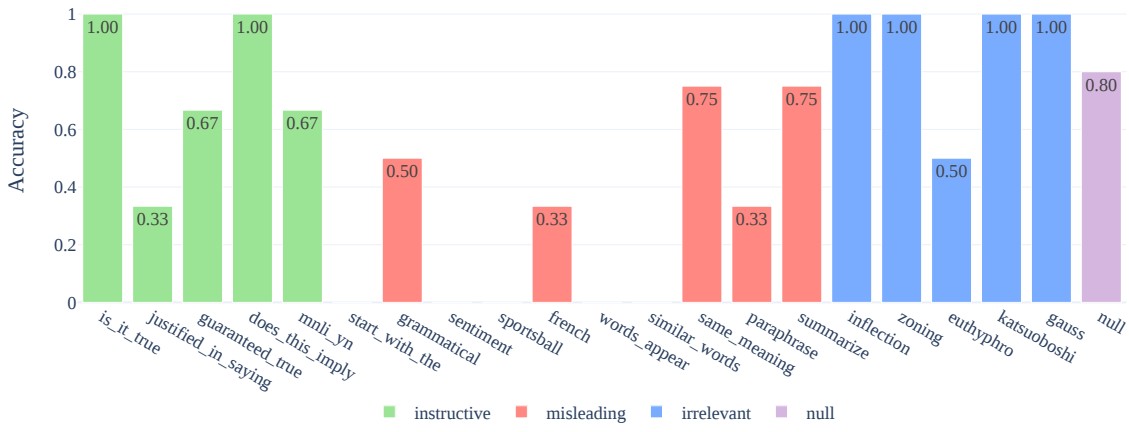

Figure 22: Per-instruction accuracy on the test condition item of subjects that were not filtered based on prior exposure to NLP tasks ($n = 67$).

# K  Few-Shot Experiment Post-Experiment Survey

| S/N | Prompt Category | How did you decide to choose "Yes" or "No"? | What did you think about the instructions we gave? |
|---|---|---|---|
| | | **Few-Shot With Labels** | |
| 1 | Mis-Moderate | In the first few questions, my strategy is to read through the entire paragraph or sentence and then decide whether the paraphrased sentence makes sense or not. However, then I started to look at the paraphrased sentence first and decide whether it is correct or wrong based on the given piece of text. Initially, I also considered whether the paraphrased sentence captured all the major details or not, but the quiz later shows that comprehensiveness is not a factor. | I'd say the instructions are not quite direct? In my opinion, it would make more send to ask if the given sentence is correct or not than to ask if it paraphrases the text. |
| 2 | Mis-Extreme | I chose my answer based on what I believed was correct. | I don't really like the question, "Is this grammatically correct". Some were definitely not grammatically correct (capitalization errors, past/present tense), but the answer was still yes. I feel like the question should be changed because it seems like the question is actually, "Is this statement true based on the context given in the paragraph". |
| | | **Few-Shot Without Labels** | |
| 3 | Irrelevant | I tried to see whether what was stated in the question was consistent with the preceding sentences. Sometimes it involved a logical deduction, and other times it was not implied at all by the other sentences but just related. Sometimes I was unsure what to choose because the premise of the question was wrong. | I was confused because that statement was included in every question, but it didn't seem relevant. |
| 4 | Mis-Moderate | I'm looking for whether the information provided in the first half can be more or less encapsulated by the second half, meaning that if one were to read the first half and another the second, they would come away to the same conclusion. | There is a level of ambiguity at first as I considered what exactly it entailed: whether or not its a "correct" statement given the context is a confounding factor, when it shouldn't influence whether or not its a good paraphrasing. |
| 5 | Mis-Extreme | I looked at whether the sentence was accurate to the information given in the text, and also if the sentence itself had correct grammatical structure. It was a little difficult because some of the sentences made inferences that weren't explicit in the given text, so I wasn't sure if that was a grammatical error or not. | Usually, I think of something as being grammatically correct when the sentence has correct grammatical structure, including punctuation and capitalization. Since most of the sentences seemed to fit this, I thought that maybe grammar also encompasses the validity of the statement based on the text, so I chose my answers based on that. |
| 6 | Mis-Extreme | I chose "Yes" when the shorter sentences present accurate information from the longer sentences. I was kind of confused about the question because most of the sentence (maybe all) seemed to be grammatically correct. | For the first two questions, I was paying attention to whether the sentences were actually grammatically correct. Later on, I just tried to see if the shorter sentences give accurate information based on the longer parag traphs above. |

Table 5: Sample of free-text responses of subjects to the questions "How did you decide to choose 'Yes' or 'No'?" and "What did you think about the instructions we gave?". Responses were elicited from subjects after they answered 32 items with their assigned prompt. Each table row indicates responses from one unique subject.