# OpenReview forum: "Are Language Models Worse than Humans at Following Prompts? It's Complicated"
_EMNLP/2023/Conference — EMNLP 2023 Findings_

### Official Review · Reviewer_k3cH · 2023-08-01

**Soundness:** 4

**Excitement:**

3: Ambivalent: It has merits (e.g., it reports state-of-the-art results, the idea is nice), but there are key weaknesses (e.g., it describes incremental work), and it can significantly benefit from another round of revision. However, I won't object to accepting it if my co-reviewers champion it.

**Paper Topic And Main Contributions:**

This paper conducts a follow-up study on the work by Webson and Pavlick (2022). W&P found that when presented with misleading prompts, LLMs perform just as well as if they were given good prompts. W&P further question whether models understand task instructions the same way as humans do. In this paper, the authors measured human performance on datasets (constructed with the same strategy that W&P used) and found that one of W&P’s assumptions do not hold for human performance. They caution future research to validate assumptions on human performance, if the research compares models with humans.

**Questions For The Authors:**

A. Why do you collapse misleading-moderate and misleading-extreme?
B. Why do you choose W&P's work to follow up?

**Reasons To Accept:**

1. Conduct empirical experiment to show that one assumption (in a previous paper) about human performance is wrong. This is a good finding because it underscores the importance of having more rigorous study before making these assumptions if one were to evaluate models in terms of “human-likeness”.
2. Sound experimental design.

**Reasons To Reject:**

1. A lot of work is built on W&P’s work, this paper mainly evaluates on human performance and provide some analysis.
2. Only follows up on W&P's work. In the introduction section, this paper identified several papers that made similar observations (i.e. models perform well when given non-ideal input). It would be nice to see how often the assumptions are violated.

**Reproducibility:**

4: Could mostly reproduce the results, but there may be some variation because of sample variance or minor variations in their interpretation of the protocol or method.

**Reviewer Confidence:**

4: Quite sure. I tried to check the important points carefully. It's unlikely, though conceivable, that I missed something that should affect my ratings.

---

> ### Author Rebuttal · Authors · 2023-08-28
>
> Thank you for your review of our paper! We agree that the scope of our study is limited in that it critiques mainly one past paper. However, we believe the current results are pithy and sufficient to show that human behavior signatures may be more complex than can be intuitively predicted, and hence by submitting this as as a short paper we hope to motivate further work to test assumptions in other studies that make claims comparing humans to models on the basis of intuited rather than observed human behavior.
>
> We collapse misleading-moderate and misleading-extreme because the overall patterns from both were the same *with respect to this paper’s hypothesis*—that models perform too well on the NLI task relative to humans when given misleading prompts. For the sake of argumentative clarity and conciseness, we decided to collapse the discussion of the two categories.

---

### Official Review · Reviewer_YaYe · 2023-08-04

**Soundness:** 4

**Excitement:**

3: Ambivalent: It has merits (e.g., it reports state-of-the-art results, the idea is nice), but there are key weaknesses (e.g., it describes incremental work), and it can significantly benefit from another round of revision. However, I won't object to accepting it if my co-reviewers champion it.

**Paper Topic And Main Contributions:**

This paper investigates the belief that humans would struggle with abnormal instructions. Their results show that humans reliably ignore irrelevant cues and perform well in tasks. However, when faced with misleading instructions, humans follow faithfully, unlike models. This result is important in cautioning against assuming uniform human behavior and underscores the importance of validating human behavior before training models.

**Reasons To Accept:**

1. Interesting subject to investigate LLMs and compare them to humans on different instruction artifacts.
2. Detailed experimentation and the result about A2 not being consistent with human behavior is interesting.
3. Contributes human data for the experiments which can be used for future works if authors make it open-source.

**Reasons To Reject:**

1. Results are interesting but limited in scope and impact. I can’t see where the findings of this paper will be useful.
2. Moreover, the only evaluated here is NLI. Evaluation on the other NLP tasks will help the paper to establish the generality of their findings.

**Reproducibility:**

3: Could reproduce the results with some difficulty. The settings of parameters are underspecified or subjectively determined; the training/evaluation data are not widely available.

**Reviewer Confidence:**

2: Willing to defend my evaluation, but it is fairly likely that I missed some details, didn't understand some central points, or can't be sure about the novelty of the work.

---

> ### Author Rebuttal · Authors · 2023-08-28
>
> Thank you for your review of our paper! We find your feedback valuable in improving the clarity of our work. To address the your following comments:
>
> We agree that the scope of our study is limited, which is why we are submitting this as a short paper. Our goal with this paper is to call attention to an interesting empirical trend that runs counter to common intuitions, and thus motivate future work on similar types of questions. In particular, we are trying to highlight the need for future work to empirically vet human behavior rather than comparing models to idealized/assumed human behavior.
>
> On scope— we agree that it would be valuable to explore tasks other than NLI. The scope is small because this is a highly controlled study that entailed significant human effort to design well. For one, the small scope is so that we can hand-pick examples for a rigorous experimental design—namely, that our examples for misleading prompts are chosen such that the answers for the explicit task and NLI task are always separable. Furthermore, NLI is the only task that was systematically held out from training by the models we tested.
>
> We certainly agree that extending the findings by validating other tasks would be a worthwhile pursuit. For this short paper submission, however, we chose to be focused and rigorous in favor of breadth, and hope that the results motivate future work in expanding to other tasks.
>
> Thank you again for your review!

---

### Official Review · Reviewer_fzxT · 2023-08-09

**Soundness:** 4

**Excitement:**

4: Strong: This paper deepens the understanding of some phenomenon or lowers the barriers to an existing research direction.

**Paper Topic And Main Contributions:**

This paper follows W&P's previous work on comparison of human and model behavior in terms of following different prompts (misleading, irrelevant, null).

The paper found that it is not necessary true in previous assumptions of human behavior through human studies.

**Reasons To Accept:**

- Interesting research question.
- Comprehensive human studies to illustrate the point.


**Reasons To Reject:**

NA

**Reproducibility:**

3: Could reproduce the results with some difficulty. The settings of parameters are underspecified or subjectively determined; the training/evaluation data are not widely available.

**Reviewer Confidence:**

2: Willing to defend my evaluation, but it is fairly likely that I missed some details, didn't understand some central points, or can't be sure about the novelty of the work.

---

> ### Author Rebuttal · Authors · 2023-08-28
>
> Thank you for your kind comments on our paper!

---

### Official Review · Reviewer_3rDv · 2023-08-12

**Soundness:** 2

**Excitement:**

2: Mediocre: This paper makes marginal contributions (vs non-contemporaneous work), so I would rather not see it in the conference.

**Paper Topic And Main Contributions:**

This paper investigates the instruction-following ability of language models and humans by comparing their performance in the NLI task under different prompt settings: instructive, misleading, and irrelevant.

Contributions: This paper empirically finds that humans and models both perform well with irrelevant instructions. In addition, when given deliberately misleading instructions, humans follow the instructions faithfully, whereas models do not.
These findings caution that future research should not idealize human behaviors as a monolith and should not train or evaluate models to mimic assumptions about these behaviors without first validating humans’ behaviors empirically.


**Questions For The Authors:**

A. Why do we need LLMs or human annotators to fulfill the task with misleading or irrelevant prompts in a zero-shot manner if the task is not clearly described at all? More examples are needed to clarify the motivation and the experimental results of this paper.

B. What does it mean if LLMs achieve higher accuracy than humans in NLI when the instruction actually guides them to do another task without any information about NLI? Isn't it natural for humans to be faithful to the instruction? Why is this phenomenon complex?


**Reasons To Accept:**

Large language models (LLMs) have shown impressive performance in various NLP tasks by following prompts.
This paper studies "whether language models are worse than humans at following prompts"  and has some findings different from previous works, which might be helpful for the LLM community to analyze the behavior of humans and LLMs.

**Reasons To Reject:**

1. LLMs are versatile in general-purpose NLP tasks, whereas this paper only conducts experiments in the NLI task using less than 200 examples (Table 2). The experimental results fluctuate too much in Figure 2, especially for misleading prompts. The result analysis and discussion are not so informative and helpful for future work.

2. The claim of this paper is a little weird. When humans are involved in evaluating NLP models, it is fundamental to describe the task clearly first. I don't understand the point of this paper, which claims "human behaviors are complex (given misleading prompts)" with few insightful conclusions.

**Reproducibility:**

4: Could mostly reproduce the results, but there may be some variation because of sample variance or minor variations in their interpretation of the protocol or method.

**Reviewer Confidence:**

2: Willing to defend my evaluation, but it is fairly likely that I missed some details, didn't understand some central points, or can't be sure about the novelty of the work.

---

> ### Author Rebuttal · Authors · 2023-08-28
>
> Thank you for the in-depth review of our paper.
>
> ## Response to criticism #1: Concerns about the scope of the study
>
> We agree that the scope of our study is limited, which is why we are submitting this as a short paper. Our goal with this paper is to call attention to an interesting empirical trend that runs counter to common intuitions, and thus motivate future work on similar types of questions. In particular, we are trying to highlight the need for future work to empirically vet human behavior rather than comparing models to idealized/assumed human behavior.
>
> The scope is small because this is a highly controlled study that entailed significant human effort to design well. We highlight that all our examples are handpicked so that the answers for an NLI task and answers for the explicitly instructed non-NLI task—in the case of misleading instructions—are always separable. We describe reasons for why the choice of NLI as our task is because it is the in-principle best task for our hypothesis in our Limitations section, and acknowledge that there is certainly room for future work in comparing human v. model instruction-following behaviors signatures on other tasks. However, we believe the current results are pithy but sufficient to show that human behavior signatures may be more complex than can be intuitively predicted, and hence hope to motivate further work by submitting this as a short paper.
>
> ## Response to criticism #2: Concerns about the claim of the paper
> Based on your question, **we believe you misunderstood the role of human subjects in our study and our overall research question:** We are trying to determine whether humans and models exhibit similar behaviors *when they are given misleading or irrelevant instructions.* This question is important to answer because there are multiple papers analyzing and criticizing models’ behavior in such a setting (see our Related Work section), but there is no data on what humans would do in comparable situations. We are *not* collecting the typical sense of a “human baseline” (which serves to judge if models perform better/worse than humans on a specific NLP task and in which case we agree informative instructions are important)
>
> To answer our research question, we not only need to collect instruction-following behavior in the best case, where “it is fundamental to describe the task clearly first”, but also in the worst cases: when (i) there are bad descriptions of the task, which are our misleading prompts, and (ii) when there are no descriptions of the task, which are our irrelevant and null prompts.
>
> Our research question also motivates why our experiments are zero-shot. We are testing instruction-following behavior, and not the task-deduction behavior which may be engaged when given few-shot examples. That is, we are giving minimal auxiliary information so as to test the intrinsic biases humans and models have in interpreting instructions.
>
> You claim that our conclusions are not insightful because it is “natural for humans to be faithful to the instruction”. **This is the kind of (intuitive) assumption that the paper is testing empirically, and in fact, finds does not always hold.** Namely, counter to this intuition, humans perform well with irrelevant instructions. The W&P paper which we are responding to also assumed this and predicted that humans would hence perform well if given instructive instructions, and badly if given misleading, irrelevant or no instructions. We show that that is not in fact what happens.
>
> Our results also show interesting subtleties in human v. model behaviors that relying on intuition alone would not reveal. For instance, while models show a strong NLI performance variance with misleading instructions based on how proximal the distractor task is to the NLI task (e.g. models are more likely to perform the NLI task when asked to do the summary task than when asked to do the grammaticality task), humans are more uniformly unlikely to perform the NLI task no matter the distractor task (compare the variance of Humans’ misleading-prompt scatter points and that of models in Figure 2).

---

### Meta-Review · Area_Chair_EP8S · 2023-09-15

**Recommendation:** 3

**Metareview:**

Pros:
- It is valuable to assess the implicit assumptions in prompt perturbation about the correct or human way to handle a misleading or irrelevant prompt. They find the "human" way to handle things is not to execute misleading instructions, but to make reasonable adjustments,  and to ignore irrelevant information. The authors argue that papers that test the consistency of prompting make this assumption often, but it has never been tested in humans.
- This paper is a direct response to a specific work on NLI, and mimics their setup, which justifies the focus on a single task.

Cons:
- Results are limited to only NLI. Not only does this make the title overly broad, but as 3rDv pointed out, the particular nature of the NLI task could easily shape the findings because it's not clear how to handle certain perturbations without direct instructions.
- Real world annotation is often performed with complex instructions, so it is hard to see whether these results would generalize in more complex settings.

---

### Decision · Program_Chairs · 2023-10-07

**Decision:**

Accept-Findings

**Comment:**

Pros:
- It is valuable to assess the implicit assumptions in prompt perturbation about the correct or human way to handle a misleading or irrelevant prompt. They find the "human" way to handle things is not to execute misleading instructions, but to make reasonable adjustments,  and to ignore irrelevant information. The authors argue that papers that test the consistency of prompting make this assumption often, but it has never been tested in humans.
- This paper is a direct response to a specific work on NLI, and mimics their setup, which justifies the focus on a single task.

Cons:
- Results are limited to only NLI. Not only does this make the title overly broad, but as 3rDv pointed out, the particular nature of the NLI task could easily shape the findings because it's not clear how to handle certain perturbations without direct instructions.
- Real world annotation is often performed with complex instructions, so it is hard to see whether these results would generalize in more complex settings.